# Zika Virus: A Review of Biology, Clinical Impacts, and Coinfections

**DOI:** 10.3390/v17050637

**Published:** 2025-04-28

**Authors:** Lucas Matheus Barreto Santana, Ingrid Andrêssa de Moura, Yuri Mouzinho Ramos Tanaka, Rafael Freitas de Oliveira França

**Affiliations:** 1Department of Virology and Experimental Therapy, Fundação Oswaldo Cruz/Fiocruz, Recife 50740-465, PE, Brazil; lucas96barreto@gmail.com; 2Laboratory of Molecular Studies and Experimental Therapy—LEMTE, Department of Genetics, Federal University of Pernambuco, Recife 50740-465, PE, Brazil; ingrid.andressa@ufpe.br; 3Department of Entomology, Fundação Oswaldo Cruz/Fiocruz, Recife 50740-465, PE, Brazil; tanakaasces@gmail.com; 4Plataforma de Pesquisa em Medicina Translacional, Fundação Oswaldo Cruz–Fiocruz São Paulo, Ribeirão Preto 14049-900, SP, Brazil

**Keywords:** pathogen interactions, flavivirus, Congenital Zika Syndrome, coinfections

## Abstract

The Zika virus (ZIKV) gained prominence as a significant global pathogen after the 2015–2016 outbreaks associated it with an increase in neurological complications in adults and congenital malformations. Different mechanisms have been proposed by which ZIKV may cross the blood–brain barrier and reach the central nervous system to cause neuroinflammation. Although ZIKV infection triggers a robust immune response, the virus has developed different strategies to escape it. Furthermore, although the virus is present in areas with cocirculation of other pathogenic agents, few studies have evaluated the cross-immune reactions and coinfection of ZIKV. Coinfections of ZIKV with other viruses, parasites, and bacteria are described. Such interactions can worsen infections and alter the immune response, imposing new therapeutic challenges and highlighting the need for more studies in the field. In this review, we discuss various aspects of ZIKV biology, focusing on the impacts of coinfections.

## 1. Introduction

In recent years, the incidence of infectious diseases affecting the central nervous system (CNS) has risen. This is due to several factors, including the emergence of new pathogens, globalization, climate change, and the increased geographic distribution of vectors [1,2]. These neuroinfections trigger a series of disorders, including exacerbated neuroinflammation, excitotoxicity, and neurodegeneration, which can result in different clinical consequences, such as cognitive and motor dysfunctions, behavioral changes, seizures, and death [3]. Among the different pathogens capable of causing neuroinfections, the Zika virus (ZIKV) has gained prominence in the last decade due to its association with severe neurological complications.

ZIKV has a pronounced affinity for neural cells and causes effects with teratogenic consequences in humans [4]. It was discovered in Uganda in 1947 [5], and its first report of human infection was in 1954 [6]. Transmission occurs mainly from the bite of an infected mosquito, with *Aedes aegypti* and *Aedes albopictus* mosquitoes recognized as the main vectors in human transmission. Sexual and vertical transmission of the virus is also recognized, in addition to reports of the presence of ZIKV in the blood of donors and in the breast milk of infected women [7,8,9,10].

ZIKV infection causes a self-limiting disease with clinical symptoms manifesting in 20% of cases. Clinical manifestations can appear up to two weeks after the infection and persist for about seven days. The most common symptoms are fever, headache, and maculopapular rash [11,12,13]. Pruritus, dizziness, anorexia, conjunctivitis, arthralgia, myalgia, stomach pain, and upper respiratory tract symptoms may also occur [14,15,16]. However, during the epidemics of 2015 and 2016 in Brazil, a link was found between ZIKV infection and a group of congenital malformations known as Congenital Zika Syndrome (CZS). Moreover, in adults, ZIKV has been associated with Guillain–Barré Syndrome (GBS) and other neurological complications, such as encephalitis, meningitis, and encephalopathy [17].

Different hypotheses have been raised for the large number of microcephaly cases caused by ZIKV infection in Brazil. Although monoinfection by ZIKV has been extensively studied, coinfection has been poorly explored [18]. These events highlighted the need for a better understanding of the virus’s biology, infection mechanisms, and interactions with other pathogens during coinfections. Therefore, this article aims to review the most recent topics on ZIKV, addressing the mechanisms of infection and immune response, interaction with the blood–brain barrier (BBB), clinical and experimental findings related to CZS, neurological manifestations in adults, and cross-immune reactions. Furthermore, since this topic remains understudied and requires further investigation, we reviewed the existing research on ZIKV coinfection with other pathogens and its potential clinical implications.

## 2. Biology of ZIKV

Two main lineages of ZIKV have been identified: African and Asian. The genetic composition and evolutionary history of these lineages vary, as does their geographic distribution [19]. The Asian lineage has a wide geographic distribution, with high prevalence in Southeast Asia and the Pacific Islands, and was responsible for the 2015–2016 epidemics in the Americas [20,21]. Unlike the Asian lineage, the African lineage has not been associated with significant or extensive epidemics. It has a restricted distribution, although it continues to circulate in some regions of Africa, being mainly present in sub-Saharan Africa [22,23].

ZIKV belongs to the *Flaviviridae* family and the flavivirus genus. It is an enveloped, positive-sense, single-stranded RNA virus with an icosahedral symmetry and a diameter of 40–60 nm. The genome has approximately 11 kb and consists of a coding region that translates a polyprotein that is flanked by two non-coding regions [24,25]. This polyprotein is processed to produce three structural proteins: capsid (C), precursor membrane (prM), and envelope (E), as well as seven non-structural (NS) proteins: NS1, NS2A, NS2B, NS3, NS4A, NS4B, and NS5 (Figure 1) [7].

The C protein binds to the genomic RNA to form the nucleocapsid. The prM and E proteins are anchored to the host cell membrane to form the viral envelope. While the seven non-structural proteins are involved in multiple stages and functions of the viral life cycle, such as RNA replication and viral particle assembly, they also participate in different immune evasion processes [26,27].

### 2.1. Infection and Immune Response

ZIKV infects a variety of cell types, such as fibroblasts, epithelial cells, dendritic cells (DCs), neurons, and trophoblasts. This varied tropism, in turn, results in a systemic ZIKV distribution, leading to its detection in tissues and body fluids, such as semen, saliva, tears, urine, ocular regions, brain, and testes [28]. Necropsies in infected fetuses have also verified the presence of the virus in the kidneys, lungs, thymus, brain, adrenal glands, and spleen [29]. Furthermore, ZIKV can bind to various cell surface receptors, including AXL, DC-SIGN, TYRO-3, and TAM-1, which aids in its entry into host cells via clathrin-mediated endocytosis [30]. Therefore, it is difficult to locate a single therapeutic target against ZIKV, due to the virus’ ability to bind to several cellular receptors in different regions of the body. For that reason, competitive inhibition of a single therapeutic target would not be sufficient to inhibit viral infection and replication.

After cell entry, ZIKV is recognized by intracellular receptors, such as Toll-like receptors (TLRs) and RIG-I-like receptors (RLRs), which detect viral RNA and activate a signaling cascade. Due to the activation, transcription factors, including NF-κβ, interferon regulatory factors (IRFs), and mitogen-activated protein kinases (MAPKs), promote the expression of pro-inflammatory and antiviral molecules, such as cytokines (IL-1β, IL-6, IL-8, and TNF-α); chemokines (CCL2 and CXCL10); and antiviral proteins (IFIT1 and RSAD2/Viperin). These molecules reduce viral replication, induce inflammation, and help immune cells be recruited to the infection site [19,31,32].

However, ZIKV has immune escape mechanisms and is capable of inhibiting IFN production by interfering with IRF3 and IRF7 [33]. The infection in human neural progenitor cells (hNPCs) and human-induced pluripotent stem cells (hiPSCs) into astrocytes are detected by RIG-I and TLR-3 receptors, but TLR3 activation by ZIKV suppresses the IFN response triggered by RIG-I receptors [34]. The viral proteins NS5, NS1, NS3, NS2a, and NS4a interact with molecules present in signaling pathways, causing the suppression of immune responses [33]. These evasion mechanisms allow ZIKV to persist in the body and modulate the immune response in a way that favors the survival of the virus. NS5 promotes the proteasomal degradation of STAT2, compromising IFN-I signaling. NS1 also inhibits NF-κB phosphorylation, while NS3 antagonizes RIG-1- and MDA5-mediated signaling. NS2a and NS4a proteins negatively regulate RLRs and downstream components of the RIG-I/MDA-5 signaling pathway [32,33].

In the developing fetal brain, infection triggers the immune response activated by microglial cells, which secrete nitric oxide, cytokines, and chemokines, thereby recruiting macrophages and natural killer cells. This pro-inflammatory milieu results in intensified neuroinflammation [32]. In neural progenitors, infection causes cell cycle and differentiation arrest while also modulating cell death pathways, leading to necrosis, paraptosis, or apoptosis [35,36]. Zika virus infection disrupts neuronal signaling pathways involving the proteins phosphatidylinositol-3-kinase (PI3K) and mTOR, which are essential for brain homeostasis [37]. Moreover, the virus induces the accumulation of misfolded proteins in the endoplasmic reticulum (ER) and alters the organelle structure for viral RNA replication, thereby activating the unfolded protein response [38]. ZIKV also elevates reactive oxygen species (ROS) in mitochondria, resulting in DNA damage that may induce apoptosis caused by the usurpation of mitochondrial metabolism to furnish energy for viral replication through the oxidative phosphorylation pathway [30]. In the Golgi complex (GC), the calcium homeostasis is also compromised during infection. Cells lacking SPCA1, which directs calcium transport to the GC, show an inhibition of ZIKV maturation, indicating that adequate calcium concentrations are essential for viral assembly [39]. Additionally, TRP (transient receptor potential) family calcium channels, such as TRPC4, are upregulated during ZIKV infection, mediated by the interaction between the viral protein NS3 and the protein kinase CaMKII. In vivo inhibition of TRPC4 and CaMKII decreased the number of seizures and increased the survival of ZIKV-infected neonatal mice, in addition to reducing the rate of cell death and viral replication in brain organoids derived from human-induced pluripotent stem cells [40].

### 2.2. Blood–Brain Barrier and ZIKV Infection

The blood–brain barrier (BBB) is an interface between the peripheral nervous system and the CNS, characterized by the presence of endothelial cells strongly adhered through occlusion junction proteins (tight junctions). These junctions are formed by structural proteins, claudins, and occludins that are associated with microglia cells, astrocytes, and pericytes [3,30]. The brain microvascular endothelial cells (BMECs) are the primary component of the BBB, lining the brain capillaries associated with other cells. BMECs are characterized by the absence of fenestrations, scant pinocytosis, and the absence of transmembrane transport molecules. It has the function of limiting the passive diffusion of polar molecules from the bloodstream to the brain, the supply of nutrients and oxygen, and the efflux of harmful metabolites and xenobiotics, maintaining the hydroelectrolytic balance and regulating the circulation of immune cells through the barrier [4,41].

There are different proposed mechanisms by which ZIKV can cross the BBB: transcellular transport within BBB endothelial cells through infection or transcytosis mediated by ZIKV-induced degradation of the Mfsd2a protein; paracellular trafficking of ZIKV across the BBB that occurs through the upregulation of pro-inflammatory cytokines, chemokines, adhesion molecules, and growth factors; downregulation of tight junction proteins; and through ZIKV-infected monocytes that cross the BBB through the Trojan horse strategy [30,42]. Once it reaches the CNS, ZIKV infects different cells, including astrocytes and microglial cells that produce cytokines and chemokines, leading to inflammation. In addition, the infection causes cell cycle disruption, mitochondrial dysfunction, ER stress, and the modulation of host autophagy and induces apoptosis, resulting in neurogenesis dysfunction, viral replication, and cell death [30].

Kaur et al. (2023) [4] performed an experimental coculture work to mimic the BBB using hBMECs and hiPSC-derived astrocytes. They observed that hBMECs transfected with the E protein had lower expression of cell junction proteins ZO-1, occludin, and VE-cadherin, leading to greater cell permeability. An upregulation of genes involved in leukocyte recruitment was also observed, along with an increase in chemokines and pro-inflammatory cytokines. Furthermore, the ZIKV envelope protein (E) also led to astrogliosis by astrocytes.

The pathophysiological mechanisms and clinical consequences of ZIKV infection, which often correlate with invasion of the central nervous system and the resulting inflammatory response, will be discussed below.

### 2.3. Congenital Zika Syndrome

ZIKV exhibits a significant affinity for human brain progenitor cells. This may explain its tendency to damage prenatal brain development, resulting in microcephaly and other neurodevelopmental abnormalities [43]. Congenital malformations due to infection can occur during any trimester of pregnancy, with the first trimester presenting the highest risk for the fetus [44,45]. Fetal abnormalities are 14 times more likely to occur when the infection happens before 11 weeks of gestation [46]. Brazil reported numerous cases of CZS during the 2015–2016 epidemics [47]. Children with CZS have epilepsy; cerebral palsy; dystonia; persistence of primitive reflexes; dysphagia; spasticity; hyperreflexia; hearing loss; and profound delays in all motor, visual, language, and cognitive domains, in addition to urinary tract disorders: urinary tract infections, neurogenic bladder, multicystic kidney disease, and kidney stones [48,49].

Autopsies of fetuses and neuroimaging examinations of microcephalic babies infected by ZIKV revealed several anomalies: diffuse cortical thinning; polymicrogyria; pachygyria; lissencephaly; hemimegaloencephaly; ventriculomegaly; hypoplasia of the corpus callosum, cerebellum, and brainstem; and calcifications in the cortex and subcortex in the white matter of the frontal, parietal, and occipital lobes and in the basal ganglia. Furthermore, optic nerve hypoplasia and macular atrophy; microphthalmia; low visual acuity regardless of abnormalities in the retina and/or optic nerve; a spectrum of pigmentary retinopathy, torpedo maculopathy, and torpedo-like vascular absence; and hemorrhagic retinopathy were also observed [50]. ZIKV inoculated intravenously and intraamniotically in pregnant Rhesus macaques caused death of one fetus after 7 days of inoculation, while the other fetuses presented ZIKV RNA in different tissues, including the brain, which exhibited calcifications and a reduction in the number of neural precursor cells [51]. ZIKV can also alter the methylation pattern of genes related to several neurological disorders in embryonic stem cell-derived organoids [52].

### 2.4. Neurological Manifestations in Adults

ZIKV has also been associated with long-term neurological sequelae in adults [53]. Nonetheless, although the neuropathology of CZS has been thoroughly examined, the neurological symptoms in adults remain little comprehended. Infection can lead to complications in the central and peripheral nervous systems [54], including transverse myelitis, encephalitis, chronic inflammatory demyelinating polyneuropathy, meningitis, seizures, and optic neuropathy. However, the most common clinical neurological manifestation is GBS [53,55]. GBS is characterized by acquired, inflammatory, demyelinating polyradiculoneuropathy. It tends to manifest a few days after the classic manifestations of the viral infection. The main clinical manifestations include the absence of reflexes, paresthesia with sensory loss, motor weakness, and albuminocytological dissociation of the cerebrospinal fluid [56].

A systematic review observed an incidence rate of 0.3% of neurological manifestations in adults infected with ZIKV. The average interval between the onset of infection symptoms and the appearance of neurological complications ranged from 5 to 10 days, with GBS accounting for 75% of the cases [53]. Characteristics linked to a higher risk of GBS included a poorer socioeconomic level and a history of prior infections. Among the confirmed cases of GBS associated with ZIKV, the mortality rate was approximately 3%. Moreover, patients with post-ZIKV GBS exhibited a higher frequency of dysphagia, paresthesias, facial weakness, and dyspnea compared to those with GBS from other etiologies [53]. On the other hand, large-scale population analyses emphasized these associations. A cohort study made by Charniga et al. (2021) [57] involved 106,000 samples of individuals with suspected or confirmed ZIKV infection during the 2015–2017 epidemics in Colombia, which revealed that pregnant women had a higher incidence of the infection compared to non-pregnant men and women. Among the recorded neurological manifestations, 85% corresponded to GBS. Additionally, a correlation was observed between advanced age and a higher risk of neurological complications, with the highest incidence in individuals aged 75 or older.

In a case report, a 29-year-old man presented with ascending paresthesia in the feet, followed by paraparesis of the lower extremities. The symptoms progressed to the trunk and extremities of the upper limbs, in addition to respiratory disorders, dysphagia, and urinary incontinence. The patient’s condition progressed to dyspnea, proximal and distal sensory loss, tetraplegia, and dysarthria with facial symmetry requiring mechanical ventilation, in addition to bilateral areflexia and decreased general tactile sensation. On day 115 after the onset of symptoms, an electromyography revealed motor and sensory polyradiculopathy, demyelination with reinnervation of the muscles of the right upper extremity, and continuous active denervation bilaterally in the lower extremities. On day 195, he felt decreased sensation and strength in his feet. Despite therapy, the patient became severely debilitated 6 months after the onset of the disease. Serological tests were positive for ZIKV and negative for dengue virus (DENV) and chikungunya virus (CHIKV) [58]. In an in vivo study, two species of nonhuman primates inoculated subcutaneously with ZIKV detected viral RNA in reproductive and neurological tissues, even after the end of viremia. This study demonstrated that ZIKV can infect the CNS and persist even in the absence of clinical signs [59].

This information highlights the severity of ZIKV infection. Although it is a self-limiting disease and, in most cases, asymptomatic, the neurological manifestations of the disease are a significant risk that can impose disabilities and death in infected patients. Curiously, even in other countries where ZIKV is prevalent, the severity of infections was especially intense in Brazil, where there were a large number of cases of microcephaly and GBS during the epidemics of 2015 and 2016. Being a tropical country with a high cocirculation of endemic agents and the presence of common vectors, among the main reasons raised were possible coinfections and cross-immune reactions.

## 3. Cross-Reactive Immunity Among Primary and Secondary Infections

One of the hypotheses suggested for the increase in cases of microcephaly associated with ZIKV infection in South America is the previous exposure of the population to other flaviviruses that may have generated cross-reactive immune responses [60]. Such reactions can result in poor viral neutralization or Antibody-Dependent Enhancement (ADE), a phenomenon in which there are no neutralizing or subneutralizing antibody levels, exacerbating inflammation or facilitating viral internalization and replication [61,62]. Despite the relevant impact on public health, studies in the role of primary infections in secondary infections are still scarce, especially in human and nonhuman primate models [63].

The in vitro analysis of the cross-reactivity potential of sera from patients previously infected with ZIKV or DENV revealed that the sera from patients infected with DENV showed cross-neutralization against ZIKV, yellow fever virus (YFV), West Nile virus (WNV), Saint Louis encephalitis virus (SLEV) and lheus virus (ILHV). On the other hand, the serum from patients infected with ZIKV neutralized ILHV, WNV, SLEV, and Rocio virus (ROCV). Furthermore, it was observed that some serum samples with antibodies against DENV blocked the entry of ZIKV into cells, while others increased the infection rate, suggesting individual variability in the immune response [61]. In another study, the follow-up of women infected with ZIKV revealed that 6 out of 10 patients had IgG antibodies against DENV [64]. In addition, Estofolene et al. (2019) [65] analyzed the influence of antibodies generated in previous ZIKV infections on the clinical impact of DENV infections. The researchers observed that individuals previously infected with ZIKV had more severe forms of DENV infection than patients previously infected with DENV. Depending on the serotype, pediatric patients in Nicaragua infected with DENV showed different associations. Individuals primarily infected with ZIKV had a higher risk of developing infection by DENV types 2, 3, and 4 but not by type 1 [66].

In vivo trials with nonhuman primates observed that primary infections with the four serotypes of DENV did not increase secondary infections with ZIKV. Similarly, primary infections with ZIKV did not affect the intensity of infections with the four serotypes of DENV [67]. Pantoja et al. (2017) [60] observed similar clinical results. However, the authors also described that prior DENV infection was able to reduce the days of ZIKV viremia, in addition to modulating the innate immune response, humoral response, and T cell immune response without increasing the infection. On the other hand, previous infections with DENV types 2 and 3 and the yellow fever vaccine promote protection against death in mice subjected to a lethal challenge with ZIKV [68].

Despite being an important subject of study for understanding the dynamics of infections in endemic regions, the number of works that have analyzed cross-reactions between primary and secondary infections remains unsatisfactory. Most reports concentrate on the interaction between infections with the ZIKV and DENV viruses. Moreover, another point that deserves attention and is also little explored in the literature is coinfections.

## 4. Coinfections

Coinfections occur when an individual presents two or more simultaneous infections, being common in the human population. These infections involve complex interactions between pathogenic agents, host cells, and the immune system, which can result in impacts on viral replication, pathogenesis, and the clinical outcome of the patient, potentially imposing additional therapeutic challenges [69,70]. However, most studies only analyze individual infections. Moreover, there are records of coinfections that were initially attributed to individual infections, indicating a possible underreporting of cases [71].

Coinfecting viruses can interact directly or indirectly. Such interactions can be competitive or synergistic and lead to changes in replication rates, gene expression, and viral evolution. During a coinfection, gene exchange can occur between different viruses from the same virus family or even between viruses of distinct families, genera, species, and strains, forming chimeric viruses. These viruses may have stronger biological characteristics, such as greater virulence and greater transmission capacity [53]. Furthermore, there are different factors that can influence the prevalence of coinfections. Different strains may coexist, have different levels of simultaneous circulation, and have the potential for coinfection. Dominant strains may be less susceptible to coinfection, competing with another pathogen within the organism. Furthermore, regions with limited availability of specific diagnostic tools may underestimate the frequency of coinfection due to misdiagnosis or inadequate reporting [69]. Such information highlights the need for more external research to understand the transmission dynamics and broader implications of arbovirus cocirculation and coinfection on human and animal health [72].

In humans, arbovirus coinfections can occur through sequential bites from mosquitoes infected with different viruses or through the bite of a vector coinfected with two or more pathogens (Figure 2) [73], while mosquitoes are coinfected during a blood meal when feeding on a coinfected individual or individuals with different infections. This problem is compounded, because arboviruses, including DENV, ZIKV, CHIKV, and YFV, are primarily transmitted by the same vector, *A. aegypti* [57]. However, a reduction in the replication rate of ZIKV in *A. aegpity* mosquitoes was observed when comparing mono- and coinfections with *DENV-2*, with no difference in the replication of DENV-2 being observed between the groups [74,75]. When compared to monoinfection, the infection rates of ZIKV also were lower in coinfections with the Mayaro virus (MAYV) [76].

In humans, different types of coinfections have already been described, such as with different subtypes of human immunodeficiency virus (HIV), different subtypes of influenza virus, different variants of SARS-CoV-2, Epstein–Barr Virus (EBV)/Human papillomavirus (HPV), HIV/herpesvirus, hepatitis B virus (HBV)/Hepatitis D virus (HDV), ZIKV/CHIKV/DENV, HIV/hepatitis C virus (HCV), HBV/HCV, and SARS-CoV-2/influenza virus [70,77]. With the geographical and temporal overlap of different pathogens, understanding the impact of possible coinfections on the pathophysiology and clinical outcome of diseases becomes important [78]. Although the clinical presentation of ZIKV monoinfections is well characterized, the impact of coinfections remains poorly understood. In subtropical and tropical regions where ZIKV is prevalent, the high incidence of other infectious diseases means that a proportion of ZIKV cases occur concurrently with one or more additional pathogenic infections [18].

### 4.1. ZIKV and DENV Coinfection: Immune Response and Clinical Outcomes

Most cases of ZIKV coinfection are with other arboviruses. However, ZIKV coinfection has been detected with several other types of pathogens (Table 1) [18]. Arboviruses are infectious agents which intermediate host is an arthropod vector, with mosquitoes being the most significant transmitters of these agents to vertebrate hosts [79,80]. Although they do not constitute a taxonomic classification, there are approximately 530 arboviruses, of which more than 100 are capable of infecting and causing disease in humans [81,82].

The most common ZIKV coinfection cases occurs with CHIKV, followed by DENV and a triple coinfection with CHIKV and DENV [18]. The four DENV serotypes, ZIKV and CHIKV viruses, share similar geographic distributions in many regions and can be simultaneously transmitted to humans by the mosquitoes *A. aegypti* and *A. albopictus* [72,73,95]. In Colombia, a triple coinfection was identified in a pregnant woman with DENV-2/CHIK/ZIKV, whose pregnancy proceeded without complications [96]. In Brazil, a study with 1289 women after a ZIKV epidemic in the state of Ceará identified 8.3% of positive cases for one or more arboviruses, with 2.3% of confirmed coinfections, including ZIKV/DENV, ZIKV/CHIKV, and ZIKV/DENV/CHIKV [77].

There have been reports of cases of ZIKV coinfection with all the four serotypes of DENV [65,83]. ZIKV and DENV-2 were isolated from a patient who traveled to Haiti and developed fever, rash, arthralgia, and conjunctivitis [97]. Six other cases of coinfection were described by Siqueira et al. (2020) [98]. During the DENV outbreaks in Sri Lanka, cases of ZIKV coinfection were also identified. The combination of exposure to ZIKV and DENV was associated with adverse outcomes during infection in adults. In one of the cases, a patient presented with severe dengue and encephalitis, and the analyses determined that the patient was simultaneously infected with DENV-2 and ZIKV [99].

The DENV/ZIKV coinfection decreases the ability of CD4+ T cells to produce IFN-γ and TNF compared to single infections by this virus [100]. In Rhesus monkeys coinfected with ZIKV and DENV-2, an increase in the number of pro-inflammatory CD14+ and CD16+ monocytes was observed, along with pro-inflammatory cytokines, despite the absence of changes in viral titers compared to single infections. Interestingly, a biphasic peak of cytokines (e.g., I-TAC, Eotaxin, RANTES, MCP-1, IFNγ, and MIG) was observed, coinciding with the differences in the replication kinetics of the different viruses. These animals also exhibited a significant decrease in the number of red blood cells and elevated levels of markers for muscle, liver, and kidney damage during the coinfection [101]. These findings suggest that DENV/ZIKV coinfection can intensify inflammatory responses and systemic pathology, potentially exacerbating disease severity even in the absence of increased viral loads.

### 4.2. ZIKV and CHIKV Coinfection

Coinfections of ZIKV and CHIKV have been reported in several countries, such as Haiti, Colombia, Brazil, Nicaragua, Ecuador, and Mexico [102]. A meta-analysis estimated a prevalence of 1% of coinfection with CHIKV in patients infected with ZIKV, with higher rates in North America (2.8%) and lower rates in Asia (0.1%) [69]. Underreporting of cases is a concern, especially in patients with prolonged rheumatological symptoms after a ZIKV diagnosis [71]. Even though they belong to different families, CHIKV and ZIKV viruses are endemic in many regions, exhibit neurotropism, and share several structural similarities [73]. The binding and entry into the host cell by ZIKV and CHIKV occur, respectively, through the binding of the viral proteins E and E2 to cellular receptors. Different common cellular receptors to which the CHIKV and ZIKV viruses can bind have been identified (DC-SIGN, hTIM-1, and TAM). Furthermore, a wide range of host proteins that interact with the proteins of both viruses was identified [102]. Demonstrating a great complexity in the interaction with the host.

Next-generation metagenomic sequencing (mNGS) of samples from 15 patients with confirmed ZIKV infection identified two cases of coinfection with CHIKV. Interestingly, the patients exhibited clinical manifestations more aligned with the virus that had the higher titer present in the samples. The first patient, with a high serum titer of CHIKV, had arthralgias that persisted for weeks, while the second patient, with the highest titer of ZIKV, presented with fever, rash, myalgia, and conjunctivitis [84]. K. R. Silva et al. (2018) [103] described a fatal case of a 30-year-old man with systemic lupus erythematosus and common variable immunodeficiency who was infected with ZIKV and CHIKV. The ZIKV RNA persisted for 275 days after the onset of symptoms. After 310 days, the patient died with their clinical condition evolving into severe arthralgia and progressive deterioration of renal function. In another case of ZIKV/CHIKV coinfection, a pregnant woman tested positive for CHIKV by RT-qPCR in both blood and placental samples, while ZIKV was detected in the fetal kidney. Fetal autopsy revealed placental and renal calcifications, along with signs of moderate-to-advanced maceration [104].

Moreover, there are reports of an association between ZIKV and CHIKV coinfection with GBS. In a case report, a patient presented various clinical manifestations, such as paresthesia, dyspnea, proximal and distal sensory loss, tetraplegia, demyelination with reinnervation of the right upper limb muscles, and active denervation bilaterally in the lower limbs. The patient presented with debilitation even after 6 months of treatment. Serological tests were positive for ZIKV (IgM, IgG, PRNT) and CHIKV (IgM). The authors describe that the symptoms and laboratory data were consistent with arbovirus-associated GBS [51]. Brito et al. (2017) [105] reported a serious case of meningoencephalitis along with acute inflammatory demyelinating polyneuropathy, which is the most common type of GBS caused by ZIKV and CHIKV coinfection, in a 74-year-old patient. Despite significant clinical improvement, the patient still presented weakness in the lower limbs after 6 months of hospitalization.

### 4.3. ZIKV and HIV-1 Coinfections

Calvet et al. (2016) [85] made the first report of a coinfection between the HIV-1 and ZIKV viruses in a patient in 2016 in Rio de Janeiro, Brazil. In this case, the patient presented only mild symptoms and had a good recovery. In Colombia, five cases of coinfection have been described. In the most severe case, a 45-year-old woman presented with facial nerve impairment, metabolic acidosis, and demyelination. Fortunately, she recovered after treatment with immunoglobulins [106]. Studies in rhesus monkeys suggested that ZIKV/HIV-1 coinfection may not significantly alter the progression of infections [107]. However, ZIKV infection in HIV-1 patients can impact the efficiency of antiretroviral therapy and increase the risk of vertical transmission. CD14+ monocytes are one of the targets of ZIKV infection, further increasing the risk of ZIKV coinfections in patients with HIV-1 [73].

### 4.4. Herpesviridae: HSV-1, EBV, HHV6, and CMV

Members of the *Herpesviridae* family, herpes simplex virus type 1 (HSV), Epstein–Barr virus (EBV), and Human Herpes Virus 6 (HHV6), have been reported in coinfections with ZIKV in different clinical cases with different outcomes. The coinfection of ZIKV with HSV-1 was reported in a case of meningoencephalitis. Based on the clinical findings, researchers hypothesize that a temporary and symptomatic ZIKV infection caused a temporary depression of the immune system, increasing a susceptibility to meningoencephalitis due to HSV-1 infection and/or a simultaneous mechanism of direct or indirect damage of both viruses to neuronal cells [86]. Cells from the trophoblast cell line Jeg 2 have reduced ZIKV viral replication and lowered the expression of inflammatory cytokines TNF-α and IL-6 when coinfected with HSV-2 [108]. A fetal necropsy revealed the simultaneous presence of ZIKV, EBV, and HHV-6 in different tissues. The presence of the ZIKV was detected in several tissues of the fetus, while the EBV was detected in the cerebral cortex and liver and HHV6 in the thymus, kidney, adrenal gland, and liver [29]. Regarding cytomegalovirus (CMV), Rosenstierne et al. (2018) [109] reported the presence of antibodies against ZIKV and CMV viruses in mothers and children with microcephaly. Despite these findings, the timing of infections could not be determined. Thus, possible cases of ZIKV and CMV coinfection during the pregnancy of women who gave birth to children with microcephaly were hypothesized.

### 4.5. Other Viruses: OROV, MAYV, B19V, and SARS-CoV-2

Other arboviruses, such as the Oropouche virus (OROV) and MAYV, have also been observed in cases of coinfection with ZIKV. Martins-Luna et al. 2020 [87] analyzed the presence of OROV, DENV, CHIKV, and ZIKV viruses in the serum of 496 patients. A total of 131 (26%) cases were positive for OROV infections. When analyzing these cases, 49 (9.9%) cases of coinfection were discovered. Of these, nine cases were OROV/ZIKV coinfections. Coinfection with MAYV has been documented in vitro using mammalian cells, where both viruses were able to infect and replicate in the same cell [76]. MAYV was identified concomitantly with ZIKV in two cases in a study involving 453 individuals suspected of infection [88].

With known cases of coinfection, other viruses, such as Erythrovirus B19 (B19V), show evolutionary parallels and exhibit symptoms related to those of ZIKV [110]. Fantinato et al. (2016) and Grayo (2021) [111,112] argued that the relationship between B19V and ZIKV may help ZIKV adapt in human hosts, increase viral genetic diversity, and cause the emergence of new pathogenicity patterns. This is especially concerning for susceptible groups, including pregnant women and newborns. In one fetus, for example, the result of this coinfection resulted in hydrops fetalis, intrauterine growth restriction, and severe anemia [113]. One study analyzed the presence of different viruses in 713 serum samples from patients with acute febrile syndrome, and 2.4% of patients tested positive for dual infection, with ZIKV being present in coinfections with CHIKV in 0.7% of cases and with B19V in 0.3% of cases. Triple infection was observed in 1.3% of cases, with 0.3% being a B19V/CHIKV/ZIKV coinfection [89].

In addition, during 2020, the simultaneous presence of SARS-CoV-2 with several arboviruses was observed, including probable cases of coinfection with ZIKV [114]. Especially in Brazil, the simultaneous circulation of SARS-CoV-2, DENV, CHIKV, and ZIKV presented a major obstacle to public health during the COVID-19 epidemic, since the symptoms of these infections can overlap, and differential diagnosis and appropriate clinical management can be difficult [115].

### 4.6. Other Pathogens: Schistosoma mansoni, Toxoplasma gondii, Leptospira spp., Plasmodium spp., and Candida tropicalis

There are few clinical and epidemiological studies on coinfections of ZIKV with bacteria and parasites. However, these interactions, especially in areas where numerous infectious diseases coexist, can worsen symptoms, complicate diagnosis, and affect the healthcare system.

In regions where toxoplasmosis is widespread, coinfection with ZIKV and other infections can have an amplified impact. A 29-year-old woman had her pregnancy clinically interrupted due to an intrauterine infection by *Toxoplasma gondii*, ZIKV, and CHIKV. PCR analysis detected the presence of all three pathogens in the amniotic fluid. The fetal imaging exams showed malformation of the central nervous system [90]. Furthermore, in environments prone to schistosomiasis, the cohabitation of an arbovirus and a helminth parasite can potentiate clinical symptoms. A notable case included a Brazilian patient who developed severe epididymitis. Laboratory analyses confirmed coinfection by ZIKV with *Schistosoma mansoni* [91]. Coinfection with *Plasmodium* spp. was documented in Nigeria in a study with 100 participants, where 15% of patients with fever were positive for malaria and seropositive for ZIKV [92].

In areas where leptospirosis is common and healthcare systems are overwhelmed, the combined power of coinfections can increase the severity of infections. After reporting the case of a patient coinfected with *Leptospira* spp. and ZIKV, Biron et al. (2016) [116] emphasized the importance of considering the possibility of coinfections in the context of the global spread of arboviruses and not just considering the differential diagnosis of acute febrile syndromes. In another case, a patient coinfected with ZIKV and *Leptospira* spp. died from renal and hepatic failure and gastrointestinal hemorrhage [93]. Finally, in an extreme case, a 30-year-old man who was rescued from the jungle with severe weight loss and abdominal symptoms died after 22 days of intensive medical treatment. Laboratory analyses detected systemic *Candida tropicalis* and Gram-negative intestinal bacterial sepsis, systemic viremia by the ZIKV, leptospirosis, and HSV-1 infection in the tongue and upper gastrointestinal tract [94].

## 5. Conclusions

In summary, ZIKV is a neurotropic flavivirus capable of causing severe outcomes, including congenital malformations and neurological complications in adults. Although it elicits a strong immune response, the virus employs multiple immune evasion strategies—such as interferon suppression and the degradation of signaling proteins—to access and persist in the central nervous system. In tropical and subtropical regions, where infectious diseases are highly prevalent, the complexity of ZIKV pathogenesis is further compounded by cross-reactive immune responses and coinfections. While some studies suggest that cross-immunity may either enhance or mitigate disease severity, findings remain inconsistent. Moreover, ZIKV can coinfect hosts alongside other pathogens, including viruses, bacteria, and parasites, potentially modifying immune responses and influencing disease progression. Reports of severe and even fatal outcomes in coinfected patients underscore the need for greater attention to these interactions. However, the current knowledge is largely based on isolated case reports. Comprehensive, systematic studies are essential to better understand how coinfections and cross-immunity shape the clinical and pathogenic landscape of ZIKV infection.

## Figures and Tables

**Figure 1 viruses-17-00637-f001:**
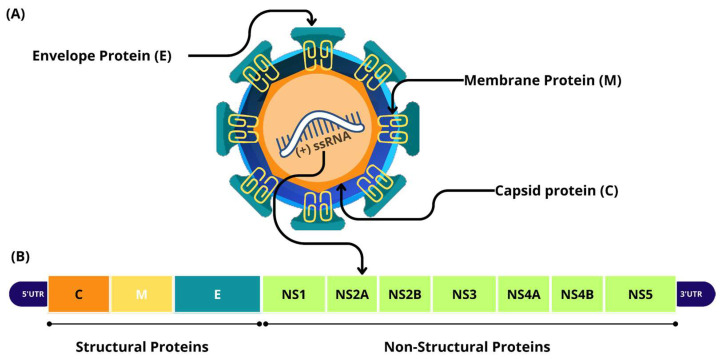
Genomic composition and viral proteins of the Zika virus. (**A**) The structure of the virion, and (**B**) the genome, which contains the sequences that encode the structural and non-structural proteins.

**Figure 2 viruses-17-00637-f002:**
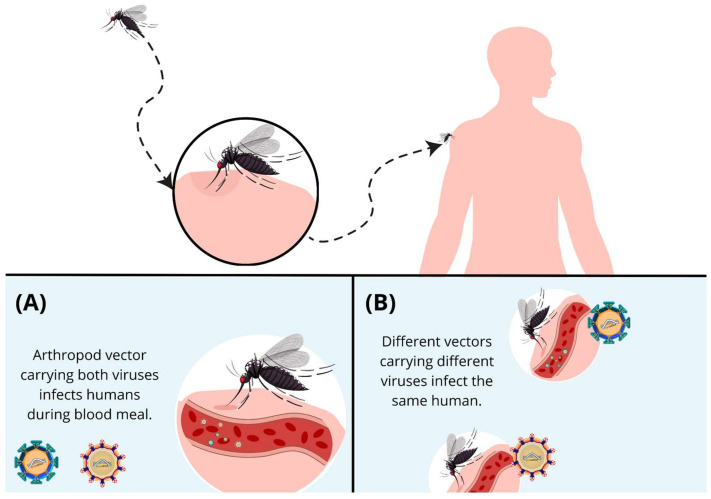
Ways in which humans can be coinfected by arboviruses. (**A**) A mosquito coinfected with two viruses feeds on an individual. (**B**) Two mosquitoes infected with different arboviruses feed on the same individual.

**Table 1 viruses-17-00637-t001:** Examples of confirmed infections between ZIKA and other pathogens.

Sample/ZIKV Detection Method	Co-Infecting Pathogen (CP)	Sample/Detection Method of the CP	References
Serum/RT-qPCR	DENV-1 and DENV-2	Serum/RT qPCR	[65]
Serum/RT-qPCR	DENV-3 and DENV-4	Serum/RT qPCR	[83]
Serum/mNGS and RT qPCR	CHIKV	Serum/mNGS and RT PCR	[84]
Serum/RT-PCR and sequencing	HIV	NI *	[85]
Serum/RT-PCR	HSV-1	Serum/RT-qPCR	[86]
Brain, thymus, lungs, kidneys, adrenal glands, and spleen/RT-qPCR	EBV	Brain and Liver/qPCR	[29]
	HHV-6	Thymus, kidneys, adrenal glands, and liver/qPCR	
Serum/RT-qPCR	OROV	Serum/RT-PCR	[87]
Serum/RT-PCR	MAYV	Serum/RT-PCR and Nested PCR	[88]
Serum/RT-qPCR	B19V	Serum/RT-qPCR	[89]
Amniotic fluid/RT-qPCR	*Toxoplasma gondii*	Amniotic fluid/PCR	[90]
Blood/RT-PCR	*Schistosoma mansoni*	Testicle/biopsy	[91]
Serum/qualitative lateral flow immuno-chromographic cassettes for IgM and IgG	*Plasmodium* spp.	Blood/Microscopy of EDTA blood sample pieces	[92]
Serum/RT-PCR	*Leptospira* spp.	Serum/PCR	[93]
Blood, liver, lung, and tissue/RT PCR	*Candida tropicalis*	Blood, pus, peritoneal and pericardial fluid/NI *	[94]

DENV: Dengue virus; CHIKV: Chikungunya virus; HIV: human immunodeficiency virus; HSV-1: herpes simplex virus 1; EBV: Epstein–Barr virus; HHV-6: Human Herpes Virus 6; OROV: Oropouche virus; MAYV: Mayaro virus; B19V: Erythrovirus B19. * NI: not informed.

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
