# Peer review of "Zika Virus: A Review of Biology, Clinical Impacts, and Coinfections"

_viruses, 2025, doi:10.3390/v17050637_

Round 1
Reviewer 1 Report
Comments and Suggestions for Authors
The manuscript describes the biology, clinical impacts and coinfections of Zika Virus. The manuscript is comprehensive but could be significantly improved. Please see comments below:
- The manuscript has many grammatical errors throughout and should be revised accordingly. Some of these are below
- Line 57 is unclear, Line 63 states SCZ which is not defines, Line 70 should read "Icosahedral Symmetry", Line 77 correct grammar.
- Lines 100 to 103 please revise to make succinct and clear to readers.
- Grammar on lines 107, 111 and 13.
- Include reference on line 114 for "IRF3 and IRF7". Reference on line 134 "for unfolded protein response". Reference on line 164 for "Trojan horse strategy"
- Line 115 is unclear.
- Line 123 define RLRs
- Section 2.3 should be divided into CZS and Adult sections. The authors have not discussed the trimester of infection and CZS. Section on adult infections is mostly 1 case report. Other larger controlled studies with statistical analysis should be discussed..
- Section 3.2. and 3.4. title shows character "e" instead of &
- Line 325 and 326 rewrite to make clear.
- Reference on line 341 and 342.
- The conclusions are largely based on case studies cited throughout the review. Authors should discuss proper case controlled studies on co infections of ZIKV with other pathogens.
Comments on the Quality of English Language
Please see comments above.
Author Response
REPLY TO REFEREES
Recife, April, 2025.
Dear Editor,
I am re-submitting the review article entitled “Zika virus: A review of biology, clinical impacts and coinfections” by Santana et al., for publication in Viruses.
All modifications are highlighted in the manuscript.
All the authors confirm that they saw and agreed to the submitted paper. The authors have been recognized as contributors and have agreed to their inclusion. The material is original, and it has been neither published elsewhere nor submitted for publication simultaneously. None of the authors has any potential financial conflict of interest related to this manuscript.
Reviewer 1
- The manuscript has many grammatical errors throughout and should be revised accordingly. Some of these are below.
- Line 57 is unclear, Line 63 states SCZ which is not defined, Line 70 should read "Icosahedral Symmetry", Line 77 corrects grammar.
- Lines 100 to 103 please revise to make succinct and clear to readers.
- Grammar on lines 107, 111 and 13.
- Section 3.2. and 3.4. title shows character "e" instead of &
- Line 325 and 326 rewrite to make it clear.
- Line 115 is unclear
- Line 123 define RLRs
Response: Dear reviewer, Thank you for highlighting the grammatical issues that required attention. Due to your feedback, we corrected the pointed observations as well as the English throughout the article to improve clarity and grammatical appropriateness. All indicated lines (57, 63, 70, 77, 100–103, 107, 111, 113, 115, 123, 325–326) have been corrected or rephrased for better readability.
- Include reference on line 114 for "IRF3 and IRF7". Reference on line 134 "for unfolded protein response". Reference on line 164 for "Trojan horse strategy"
R: We have added the requested references. For IRF3/IRF7 we added citation to Serman & Gack (2019). Unfolded protein response: Added citation to Mufrith et al. (2021). Trojan horse strategy: Added citation to Ayala-Nunez & Gaudin (2020).
- Section 2.3 should be divided into CZS and Adult sections. The authors have not discussed the trimester of infection and CZS. Section on adult infections is mostly 1 case report. Other larger controlled studies with statistical analysis should be discussed.
R: We agree with this suggestion. Section 2.3 has now been restructured into two sections:
2.3 Congenital Zika Syndrome (CZS): Expanded with discussion on trimester-specific risks (e.g., higher risk in first trimester) and population-based studies (e.g., Shapiro-Mendoza et al., 2021).
2.4 Neurological Manifestations in Adults: Now includes systematic reviews (Halani et al., 2021) and cohort studies (Charniga et al., 2021) beyond case reports.
- The conclusions are largely based on case studies cited throughout the review. Authors should discuss proper case controlled studies on co-infections of ZIKV with other pathogens.
R: We acknowledge this limitation. The conclusion now explicitly discusses the scarcity of controlled studies and emphasizes the need for further research (e.g., cohort or in vivo studies comparing mono- vs. co-infections).
“ZIKV is a flavivirus that can cause congenital malformations and severe neurological manifestations in adults. Despite triggering a robust immune response, the virus has different mechanisms that promote immune escape and allow it to reach the CNS, including the interference in interferon expression and degradation of signaling proteins. Furthermore, especially in tropical and subtropical regions, where many diseases are easily disseminated, cross-immune reactions and co-infections can increase the complexity of ZIKV infections. Although still scarce, some studies indicate that cross-immune reactions can alter the severity of infections involving ZIKV positively or negatively, although other studies have not observed such alteration. In addition, the virus has the ability to co-infect humans with other pathogens (viruses, bacteria, parasites, etc.). As a result, these interactions can alter the immune response, impact pathogenicity, and influence the severity of infections. For example, case reports have described the death of patients co-infected with CHIKV or EBV and HHB-6. In addition, a severe case of infection with neurological manifestations has also been described in patients co-infected with ZIKV and DENV-2. However, most studies are case reports. Therefore, more studies are needed to demonstrate differences in pathogenesis and clinical development between monoinfections and coinfections..”
Reviewer 2 Report
Comments and Suggestions for Authors
ZIKV infection triggers a robust inflammatory innate immune response and employs immune evasion strategies that enable the virus to cross the blood-brain barrier, causing severe neurological dysfunction. In this manuscript, Santana et al. review ZIKV biology, transmission across the blood-brain barrier, and the clinical impact of co-infections. Given that Aedes mosquitoes transmit multiple arboviruses (YFV, CHIKV, DENV, and ZIKV) and that epidemic regions overlap, the risk of co-infection is high. While the clinical consequences of ZIKV mono-infections are well characterized, the impact of co-infections in endemic areas remains unclear. Coinfections and sequential infections can modulate the host immune response, affecting disease outcomes. However, the influence of ZIKV co-infections on secondary infections and clinical manifestations remains underreported.
Overall, this review provides updated insights into the clinical outcomes of ZIKV infection, particularly the spectrum of congenital Zika syndrome (CZS)-associated neurodevelopmental disorders. It highlights the effects of co-infection with other arboviruses, chronic viral infections (such as HIV and CMV), and parasitic infections. By elucidating virus-virus and virus-host interactions, this review highlights the importance of the need for targeted prevention and treatment strategies for both single and co-infections with ZIKV and other pathogens
In the manuscript, the authors described the influence of prior infection on the immune response to the secondary infection and clinical consequences of coinfection in arbitrary ways, making it difficult for readers to follow that there are consistent patterns in the host immune response or the severity of diseases. It would been helpful to provide a table summarizing the impact of immune response (exacerbation or suppression) and the spectrum of clinical symptoms in ZIKV-infected pregnant and nonpregnant patients with coinfection based on pathogens instead of the current Table 2.
Additional minor points:
L62, a typo in SCZ for CZS.
L171, add citation [4].
P7, Table 2: The column header/title section should be on the same page as the majority of the table content.
I do not see the significance of listing virus detection methods. Instead, a list of the clinical impact and severity of co-infections based on pathogens would be more insightful, particularly triple+ infections among arboviruses (e.g., DENV/CHIKV/ZIKV). Additionally, I wonder whether the order of infection influences clinical outcomes or disease severity. It would be worthwhile to summarize in a separate table.
L262-263: The statement, “A. aegypti mosquitoes… had increased virus production and vector susceptibility to infection [58]” is incorrect. Peng et al. (58) reported that, regardless of the order of viral infection in sequential infections, prior infection with one arbovirus significantly reduces susceptibility to secondary infection, viral replication, and transmission dynamics. Please revise accordingly.
L 317, 3.2. title to be ZIKV “and” CHIKV coinfection, instead of “e”
L 369, …HHV6 “e” CMV
L350, a typo in IgF
Author Response
REPLY TO REFEREES
Recife, April, 2025.
Dear Editor,
I am re-submitting the review article entitled “Zika virus: A review of biology, clinical impacts and coinfections” by Santana et al., for publication in Viruses.
All modifications are highlighted in the manuscript.
All the authors confirm that they saw and agreed to the submitted paper. The authors have been recognized as contributors and have agreed to their inclusion. The material is original, and it has been neither published elsewhere nor submitted for publication simultaneously. None of the authors has any potential financial conflict of interest related to this manuscript.
Reviewer 2
- In the manuscript, the authors described the influence of prior infection on the immune response to the secondary infection and clinical consequences of coinfection in arbitrary ways, making it difficult for readers to follow that there are consistent patterns in the host immune response or the severity of diseases. It would be helpful to provide a table summarizing the impact of immune response (exacerbation or suppression) and the spectrum of clinical symptoms in ZIKV-infected pregnant and nonpregnant patients with coinfection based on pathogens instead of the current Table 2.
R: We thank the reviewer for the constructive feedback, which has strengthened the manuscript. While we could not include a detailed coinfection table due to data limitations, we have synthesized key clinical patterns textually. We agree that a comparative table would be valuable. However, due to the lack of standardized clinical data (most coinfection reports are case studies).
- Additional minor points:
- L62, a typo in SCZ for CZS.
R: The acronym has been corrected.
- L171, add citation [4].
R: The citation was added.
- P7, Table 2: The column header/title section should be on the same page as the majority of the table content.
R: Table title text has been adjusted.
- I do not see the significance of listing virus detection methods. Instead, a list of the clinical impact and severity of co-infections based on pathogens would be more insightful, particularly triple+ infections among arboviruses (e.g., DENV/CHIKV/ZIKV). Additionally, I wonder whether the order of infection influences clinical outcomes or disease severity. It would be worthwhile to summarize in a separate table.
R: Different studies have detected ZIKV in different regions of the body, which reflects the versatility of the virus infection. Therefore, we believe it is appropriate to keep the table in the text to demonstrate the importance of a more complete approach to detecting the virus. Regarding the order of infection, we have added a section called "Cross-reactive immunity among primary and secondary infections" describing this situation. However, there are few studies that have shown the impact of the order on the severity of the infection, and they are not sufficient for a table because they are primarily the result of Dengue and Zika infections.
- L262-263: The statement, “A. aegypti mosquitoes… had increased virus production and vector susceptibility to infection [58]” is incorrect. Peng et al. (58) reported that, regardless of the order of viral infection in sequential infections, prior infection with one arbovirus significantly reduces susceptibility to secondary infection, viral replication, and transmission dynamics. Please revise accordingly.
R: Thank you for pointing out this error, the information has been duly corrected.
- L 317, 3.2. title to be ZIKV “and” CHIKV coinfection, instead of “e”
R: We corrected the grammatical error.
- L 369, …HHV6 “e” CMV
R: We corrected the grammatical error.
- L350, a typo in IgF
R: We corrected the grammatical error.
Round 2
Reviewer 1 Report
Comments and Suggestions for Authors
Line 368 replace "no neutralizing" with "non neutralizing" antibodies.